# HLA-DRB1 Alleles Associated with Lower Leishmaniasis Susceptibility Share Common Amino Acid Polymorphisms and Epitope Binding Repertoires

**DOI:** 10.3390/vaccines9030270

**Published:** 2021-03-17

**Authors:** Nicky de Vrij, Pieter Meysman, Sofie Gielis, Wim Adriaensen, Kris Laukens, Bart Cuypers

**Affiliations:** 1Department of Computer Science, University of Antwerp, 2020 Antwerp, Belgium; Nicky.deVrij@uantwerpen.be (N.d.V.); pieter.meysman@uantwerpen.be (P.M.); sofie.gielis@uantwerpen.be (S.G.); 2Department of Clinical Sciences, Institute of Tropical Medicine, 2000 Antwerp, Belgium; wadriaensen@itg.be; 3Antwerp Unit for Data Analysis and Computation in Immunology and Sequencing (AUDACIS), University of Antwerp, 2020 Antwerp, Belgium; 4Biomedical Informatics Network Antwerpen (Biomina), University of Antwerp, 2020 Antwerp, Belgium; 5Department of Biomedical Sciences, Institute of Tropical Medicine, 2000 Antwerp, Belgium

**Keywords:** *Leishmania*, immunoinformatics, antigen presentation, HLA association, leishmaniasis, vaccine candidates, immunogenetics, human leukocyte antigen

## Abstract

Susceptibility for leishmaniasis is largely dependent on host genetic and immune factors. Despite the previously described association of human leukocyte antigen (HLA) gene cluster variants as genetic susceptibility factors for leishmaniasis, little is known regarding the mechanisms that underpin these associations. To better understand this underlying functionality, we first collected all known leishmaniasis-associated HLA variants in a thorough literature review. Next, we aligned and compared the protection- and risk-associated HLA-DRB1 allele sequences. This identified several amino acid polymorphisms that distinguish protection- from risk-associated HLA-DRB1 alleles. Subsequently, T cell epitope binding predictions were carried out across these alleles to map the impact of these polymorphisms on the epitope binding repertoires. For these predictions, we used epitopes derived from entire proteomes of multiple *Leishmania* species. Epitopes binding to protection-associated HLA-DRB1 alleles shared common binding core motifs, mapping to the identified HLA-DRB1 amino acid polymorphisms. These results strongly suggest that HLA polymorphism, resulting in differential antigen presentation, affects the association between HLA and leishmaniasis disease development. Finally, we established a valuable open-access resource of putative epitopes. A set of 14 HLA-unrestricted strong-binding epitopes, conserved across species, was prioritized for further epitope discovery in the search for novel subunit-based vaccines.

## 1. Introduction

Leishmaniasis is a vector-borne disease transmitted through sand flies and caused by protozoan parasites of the genus *Leishmania*. Leishmaniasis imposes a substantial global burden, affecting 1.5 million people annually [1]. It is one of the 20 “neglected tropical diseases”, predominantly affecting the poorest populations in low- to middle-income countries [2]. Leishmaniasis is endemic in 87 countries, spanning multiple continents and predominantly affecting countries with a warm and tropical climate. Indeed, Afghanistan, Algeria, Brazil, Colombia, Pakistan, and the Syrian Arab Republic represent more than 70% of all cutaneous leishmaniasis (CL) cases globally. Brazil, India, South Sudan, and Sudan host 78% of all visceral leishmaniasis (VL) cases. However, due to climate change, the geographical distribution might expand towards cooler regions [3,4,5].

Leishmaniasis features a wide spectrum of clinical manifestations, ranging from the skin disease cutaneous leishmaniasis to visceral leishmaniasis, which affects the visceral organs and is lethal without treatment. CL clinical presentations can also range from the often self-healing localized CL (LCL) to diffuse CL (DCL) which imposes numerous diffuse skin lesions, and mucocutaneous CL (MCL) where infected individuals develop disfiguring mucosal lesions. CL can be caused by around ten distinct species of *Leishmania*, including *L. major*, *L. braziliensis,* and *L. mexicana*, while VL is predominantly caused by *L. donovani* and *L. infantum* [6]. The majority of infected individuals do not develop symptomatic disease [7,8,9]. Yet, knowledge concerning why some infected individuals remain asymptomatic while others develop disease is scarce. The development of the different clinical manifestations is primarily dependent on the infecting species of *Leishmania*, but genetic factors and the immune response of the host play a central role as well [10].

Several major histocompatibility complex (MHC) class I and class II genes of the human leukocyte antigen (HLA) gene cluster have been identified as genetic susceptibility factors for leishmaniasis, with variants affecting disease outcome both positively and negatively [11,12]. The HLA gene complex is located within the human chromosome 6p21 region and is highly polymorphic [13]. An individual carries several HLA gene variants, or alleles, on a chromosome. The entirety of an individual’s HLA genes on a single chromosome is known as a haplotype. The varying HLA genes code for distinct MHC glycoproteins, divided into class I and class II, and these MHC molecules present antigens to CD8^+^ and CD4^+^ T cells, respectively, eliciting a T cell-mediated immune response upon activation. This T cell-mediated immunity dictates the antileishmanial immune response in both animal models and human disease [14]. However, findings in vitro or in animal models may not translate to protective or detrimental responses in humans [14].

Despite the previous identification of specific genes of the HLA loci as genetic susceptibility factors, little is known about the properties (e.g., common amino acid variants and their physicochemical properties, binding affinity and epitope dominance hierarchy) that underpin these associations. The identification of HLA variants as genetic susceptibility factors and the importance of T cell-mediated immunity suggests that antigen presentation plays a pivotal role in disease development and progression. Yet, several complications hamper research into antigen presentation in leishmaniasis. Firstly, *Leishmania* is a eukaryotic parasite with complex cellular mechanisms, such as immune modulation, and several life stages. Secondly, *Leishmania* exhibits a multitude of possible epitopes due to its relatively large proteome, coding for roughly 8000 proteins. As a consequence, relatively few epitopes have been characterized for murine leishmaniasis and even less so for human leishmaniasis [15]. Yet, identifying and characterizing these epitopes and their binding properties are crucial to understanding and studying the complex host–pathogen interaction process, but also to guide the development of safe and effective multi-epitope-based vaccines.

Traditionally, epitope discovery was centered around long and low-throughput epitope screenings. In recent times, immunoinformatics-based T cell epitope predictions, using computational prediction algorithms, allow for high-throughput in silico epitope identification and prioritization [16]. These prediction tools have thus enabled reverse-vaccinology approaches that can expedite traditional epitope-based vaccine discovery. However, the usability of these immunoinformatic predictions is not limited to reverse-vaccinology, as these tools have been used to uncover links between disease pathogenesis and antigen presentation as well [17]. In *Leishmania* research, these strategies have been widely used to predict potential vaccine candidates [18,19,20,21]. As most of these approaches require relatively high computing power, they started from only a subset of *Leishmania* proteins, mostly limited to those that are known to be immunogenic in animal models. This strategy is, however, hampered by the observed discordance between animal models and humans [14]. In addition, these immunoinformatic tools have not been fully wielded to flesh out associations between leishmaniasis pathogenesis and antigen presentation.

Therefore, we performed unprecedented multi-species and proteome-wide T cell epitope binding predictions to (A) explore the properties underpinning the known class I and class II HLA associations in human leishmaniasis and (B) establish a comprehensive resource of promising epitopes linked to broad-class protection and risk for both CL and VL. Furthermore, we provide an exploratory analysis of the physicochemical and binding properties underpinning the observed variation in protection and risk-associated HLA profiles.

## 2. Materials and Methods

### 2.1. Literature Search for Leishmaniasis-Associated HLA Alleles

We conducted a literature review searching for HLA alleles that are significantly associated with genetic susceptibility for any type of leishmaniasis. Specifically, PubMed and Web of Science were screened for relevant publications using the following keywords in all fields: ((“*Leishmania*” OR “Leishmaniasis” or “Kala-azar”) AND (“HLA” OR “Human leukocyte antigen” OR “Histocompatibility”)). EndNote X9 was used to filter out any duplicate publications found in the resulting list. The abstracts of the resulting publications were screened for relevance, and only relevant studies on human hosts were included for review. Following this, a complete screening of the full texts was performed to identify publications suitable for inclusion. HLA alleles associated with genetic susceptibility to leishmaniasis were extracted from the resulting publications. The identified HLA alleles were labeled as either ‘Risk’ for susceptibility-associated loci or ‘Protection’ for protection-associated loci (as defined by authors). The demographic information of the human study populations was also collected. The PRISMA statement guidelines were followed for article selection [22].

### 2.2. Multiple Sequence Alignment of Leishmaniasis-Associated HLA Alleles

The protein sequences of leishmaniasis-associated HLA-DRB1 alleles were acquired through the IPD-IMGT/HLA database (release 3.42.0, 15 October 2020) [13]. The HLA-DRB1 locus was specifically chosen for a sequence-based comparison, in order to unravel common amino acid variants that underpin the disease association status of an allele. The diverse spectrum of leishmaniasis-associated alleles in the HLA-DRB1 locus enabled us to filter out naturally occurring variation in the amino acid sequences, thus allowing identification of substitutions likely to be causal. The first 29 amino acid residues were trimmed off to fit a position numbering system that confers structural equivalence across the different alleles [23]. This approach allows the identification of functionally important positions, such as peptide-binding or T cell receptor contact positions. The resulting trimmed sequences were aligned using Clustal Omega with default parameters, and subsequently exported to Jalview (version 2.11.1.3) for visualisation [24,25]. Amino acids were coloured using the Zappo colour scheme, which describes their physicochemical properties and colours only non-identical positions.

### 2.3. Prediction of Epitope Binding to Leishmaniasis-Associated HLA Alleles

The proteomes of the different *Leishmania* species with known HLA allele associations were downloaded from TriTrypDB (release 46), except for *L. donovani*, for which we used the PacBio assembly by Dumetz et al. [26,27]. Pseudogenes were excluded because they are unlikely to be translated into proteins, and thus to be processed into epitopes. Epitope binding predictions across protection- and risk-associated HLA alleles were restricted to the *Leishmania* species known to be associated with a particular HLA allele.

Epitope binding prediction requires specific 4-digit resolution HLA alleles as input. Thus, in case only a 2-digit allele group of HLA was identified in the literature review, the “classical allele freq” search tool of the Allele Frequencies Net Database (accessed on 11 February 2020) was used to find the most common alleles in the population studied [28]. All alleles were subsequently cross-checked with the extensive list of available HLA proteins included in the tools to narrow down the chosen allele sequences to only those for which predictions were possible using the chosen tools.

Proteome-wide prediction of epitope binding to leishmaniasis-associated HLA alleles was carried out using the NetMHCpan v4.1b and NetMHCIIpan v4.0 tools [29]. The NetMHCpan peptide fragment length was set to 9 mer, which deviates from the default range of fragment lengths (between 8–11 mer). For NetMHCIIpan the default peptide fragment length of 15 mer was chosen. For both tools, a sliding window of 1 was used. 9 mer and 15 mer fragment length peptides are the most commonly eluted ligands for MHC-I and MHC-II, respectively [30,31]. Predicted epitopes, their binding cores, their binding affinity (BA) to a specific HLA allele, the eluted ligand score percentage rank (%Rank), and the protein(s) each epitope is derived from were extracted from the output of the epitope binding prediction tools. The %Rank is the percentile rank of the epitopes, as defined by the probability of it being an eluted ligand when compared to a set of 100,000 random naturally occurring peptides.

### 2.4. Standardization of Epitope Binding Affinity

The predicted affinity values should be standardized according to a common reference point to enable the comparison of binding preference across alleles, analogous to the method described by Meysman et al. [17]. In brief, a collection of 107 highly expressed human proteins was used to calculate the relative binding affinity of each epitope to a particular HLA allele. The binding affinity of the best scoring predicted epitope of the human dataset was then divided by the binding affinity of a predicted epitope of *Leishmania* spp. to determine a relative affinity for that particular *Leishmania* spp. epitope. This is based on the hypothesis that human proteins are the most likely competitors for binding to HLA alleles and are thus best suited as reference points.

To screen the resulting predicted epitopes for those with a maximal likelihood of being presented in vivo, two filters were applied. Firstly, only leishmanial epitopes with a %Rank lower than or equal to 2 were considered to be suitable for further analysis. Secondly, only those leishmanial epitopes with a relative score above 1 (i.e., those that bind better than the best binding human epitope) were considered likely to outperform competitor self-peptides and thus get presented. Predicted epitopes that pass both these criteria are further referred to as strong-binding epitopes.

*Leishmania* genomes feature a lot of tandemly repeated genes that are known to vary between strains of the same species [32]. Together with conserved gene families, this can lead to the same epitopes being included in many proteins identical in sequence and function, varying only in gene identifier. For these duplicate epitopes, only one was included for further analysis for each HLA allele.

The described standardization and filtering steps have been integrated into a bioinformatic pipeline, available online: https://github.com/BioinformaNicks/LeisHLA (accessed 17 Febuary 2021).

### 2.5. Characterization of Epitope Binding Repertoires

Firstly, the relative affinity distributions of all epitopes for each HLA allele were plotted on a protection-associated against risk-associated basis. Subsequently, linear mixed models were implemented, using the lmerTest package in the programming language R, to identify differences in the relative affinity distributions of protection- and risk-associated HLA alleles [33]. Here, the HLA association status served as a fixed effect while the specific HLA alleles served as a random effect. This model enables us to explore the difference between disease association status of the HLA alleles, while the intrinsic variability between HLA alleles is controlled for.

Finally, common amino acid variants within the sequences of the different leishmaniasis-associated HLA profiles might translate to functional variability within the peptide-binding groove. In order to explore this, the in silico predicted epitopes were directly wheeled for the creation of consensus sequence logos of the 9 mer binding cores. The predicted 9 mer binding cores were used to create these consensus sequence logos for each HLA allele, using the ggseqlogo R package [34].

### 2.6. Epitope Prioritization for Future Discovery

Potential strong-binding epitopes were prioritized based on two features. The overlap of these different properties was used to conclude a prioritized list of potential epitopes that are most suitable candidates for future epitope discovery.

The first feature is the conservation of predicted epitopes between species. Epitopes that are conserved between different species are more interesting candidates as they (1) increase the coverage of a vaccine therefore protecting against multiple species, and (2) are less likely to mutate over the course of evolution as conserved sequences tend to be part of essential functional domains of a protein.

The second feature of interest is the promiscuity of epitopes. Promiscuous epitopes that bind across different alleles are more suitable for discovery because they tend to be immunodominant epitopes towards which the bulk of the immune response is skewed [35]. Moreover, this prevents the confounding of immunological evaluation by HLA type restrictions.

## 3. Results

### 3.1. HLA Alleles Associated with Protection or Risk

Following the outlined search strategy of leishmaniasis-associated HLA literature, 752 publications were identified for further screening. After duplicate filtering, 451 were retained, and 33 were selected after title and abstract screening. Based on the full texts and predefined exclusion criteria, 16 were deemed highly relevant for inclusion. Seven publications were further excluded because they contained either no associations or non-significant associations after multiple testing correction. This workflow is outlined in the PRISMA Flow Diagram (Appendix A).

The literature review resulted in a list of HLA alleles that were either associated with increased susceptibility or protection for infection by *Leishmania* spp. (Table 1). This list consists of 12 high-resolution 4-digit alleles, 5 broad haplotypes, and 11 low-resolution 2-digit alleles. Most association studies conducted case/control frequency enrichment analysis to define protection- or risk-associated alleles. However, Fakiola et al. and Lara et al. performed a familial segregation analysis in addition to a case/control frequency enrichment, improving statistical confidence [36,37,38]. Protection-associated alleles were generally defined as alleles that occur significantly more in healthy endemic controls than in patients with diagnosed clinical symptoms, and vice versa for risk-associated alleles. It is important to note that several association studies selectively typed only either class I or class II HLA.

The associations in Table 1 were cross-checked with a list of HLA alleles available in the chosen prediction tools. Consequently, a subset of these HLA alleles or haplotypes were excluded from analysis due to unavailability, namely: Bw22-CF31, Bw22-DR*11-DQ*7, DRB1*0407-DQA1*3011-DQB1*0301, DRB1*0407-DQA1*3011-DQB1*0302, and DRB1*15-DQB1*06. Moreover, the HLA allele groups A*28, B*5, Bw22, and DQw3 are ‘broad antigen serotypes’, defined as a category of allele groups that are difficult to discern from each other with older serotyping methods [46]. Although newer techniques have enabled the distinction of the ‘split antigens’ within these broad antigen serotypes, it is hard to distinguish which split antigen was truly associated with leishmaniasis at the time several of these studies were carried out. Therefore, these broad antigens and associated haplotypes (Bw22-DR*11-DQ7 and Bw22-CF31) were excluded to minimize noise. The remaining associated haplotypes were also excluded for predictions due to unavailability in the prediction tools. Concludingly, the list of HLA alleles included for further analysis was narrowed down to 12 high-resolution class II HLA alleles, two low-resolution class II HLA alleles, and five low-resolution class I HLA alleles.

### 3.2. Sequence-Based Comparison of Leishmaniasis-Associated HLA-DRB1 Alleles

Only a small number of HLA class I alleles and an even smaller number of non-HLA-DRB1 class II alleles are associated with leishmaniasis. Due to this small number, variation in the sequences of these alleles is not a reliable proxy to infer the correlation of amino acid substitutions with protection-or risk-associated effects. In contrast, HLA-DRB1 alleles are widely associated with leishmaniasis. Thus, the amino acid sequences of the different leishmaniasis-associated HLA-DRB1 alleles were aligned to identify distinct substitutions in the HLA profile that explain the disease association status. This multiple sequence alignment, shown in Figure 1, identified several key amino acid substitutions that are completely (*n* = 3) or largely (*n* = 10) shared in either protection- or risk-associated HLA alleles. Interestingly, some of these are located in the functionally important HLA-DRB1 binding pocket at positions 1, 4, 6, and 9 [23,47,48,49]. These residues and their functional positions are shown in Table 2.

The leishmaniasis protection- and risk-associated HLA-DRB1 alleles can be completely distinguished at residues 9, 37, and 140. Indeed, the protection-associated alleles share a tryptophan at residue 9, a serine on residue 37 and an alanine at residue 140. In sharp contrast, the risk-associated alleles share a glutamic acid at residue 9, a tyrosine at residue 140 and exhibit a range of different amino acids at residue 37. These substitutions confer distinct physicochemical properties on these functionally important residues. The shared tryptophan at residue 9 of the protection-associated alleles is an aromatic and hydrophobic amino acid, while the glutamatic acid, shared across the risk alleles, is negatively charged. Similarly, the serine at residue 37 of the protection-associated alleles confers a hydrophilic characteristic. The risk alleles exhibit aromatic amino acids at this position, with the exception of DRB1*13:01 and DRB13:02 containing the hydrophilic asparagine. Finally, a hydrophobic alanine is shared across residue 140 of the protection-associated alleles, while the risk alleles share a hydrophilic threonine. Residue 140 is postulated to interact with the CD4 co-receptor, and the distinct physicochemical properties at this residue may influence CD4 co-receptor binding and subsequent TCR-pMHC interactions [23]. Consequently, the identified amino acid substitutions might alter the binding affinity and stability of a peptide by only enabling specific amino acids to bind at these binding pocket positions.

### 3.3. Prediction of Binding Epitopes and Their Relative Affinity

To explore the potential functional effects of the identified variants, and to provide a valuable resource of epitopes for future *Leishmania* immune research, binding epitopes were predicted across the above-defined protection- and risk-associated HLA profiles and the species corresponding to these alleles.

Predictions of 9 mer HLA class I epitopes were performed for the complete *L. braziliensis*, *L. donovani,* and *L. major* proteomes. Using a sliding window of 1 in the NetMHCpan tool, a total of 4,580,567, 5,226,608, and 5,326,280 HLA class I 9 mer peptides were respectively derived from these proteomes. Even though HLA associations were also identified for *L. guyanensis*, this species was excluded from HLA class I predictions due to the unavailability of a high-quality reference proteome. With regards to HLA class II epitope predictions, 15 mer peptides were derived from the proteomes of *L. braziliensis*, *L. donovani*, *L. infantum,* and *L. mexicana*. A total of 4,535,501, 5,175,104, 5,144,590, and 4,996,408 HLA class II 15 mer epitopes were derived from the respective proteomes, using a sliding window of 1 in the NetMHCIIpan tool.

We performed a series of subsequent filtering steps to narrow down these peptides to only those strong-binding epitopes that are likely to get presented in vivo with sufficiently high affinity to displace competitor self-peptides (see method section). Moreover, only those unique to each HLA allele (i.e., contained only one epitope of tandemly repeated gene duplicates), were considered strong-binding epitopes. The resulting strong-binding epitopes and their properties are listed in a spreadsheet provided at https://github.com/BioinformaNicks/LeisHLA (accessed 17 February 2021).

The relative affinity distributions of the strong-binding HLA class II epitopes were plotted for each species (Appendix A).

Using linear mixed effect models, no significant difference (*p* < 0.05) was observed between the relative affinity distributions of protection- and risk-associated alleles. Therefore, a general alteration in binding affinity for *Leishmania* epitopes conferred by amino acid substitutions in the HLA-DRB1 profile probably does not explain the differences between the protection or risk status of the associated HLA alleles.

### 3.4. Relationship between Amino Acid Substitutions in the HLA Repertoire and Epitope Binding Cores

Since the strength or affinity of the peptide-MHC-II binding complex is not correlated with the disease association status, we hypothesize that the amino acid substitutions across the HLA-DRB1 profiles might alter the epitope binding repertoires of the different HLA alleles. To explore this hypothesis, we visualized the binding cores of strong-binding epitopes, using consensus sequence logos to identify binding motifs within the protection- or risk-associated HLA-II alleles.

Interestingly, epitopes that bind to protection-associated class II HLA-DRB1 alleles exhibit a common combination of motifs at binding core positions 4, 6, and 9 (Figure 2). In sharp contrast, this combination of motifs is not seen in risk-associated HLA-DRB1 alleles, which display a more promiscuous nature across the binding core positions. These findings overlap with the previously identified amino acid substitutions that correspond to these binding core positions in the protection- and risk-associated HLA profiles, confirming a relationship between these variants and the (predicted) epitope binding repertoires.

Positions 4 and 9 across protection-associated DRB1 alleles consist predominantly of similar hydrophobic and polar amino acids, while position 6 consists mainly of the SA(N)G motif. While the risk-associated DRB1*13:02 shares the combination of a SANG motif at position 6 and a hydrophobic motif at position 9, it does not share the hydrophobic motif on position 4. Within the protection-associated alleles, position 4 displays a unique Y motif for DRB1*15:01 and DRB1*15:02, which maps to an R71A amino acid substitution. This R71A amino acid variant, however, subsequently maps to the LAMI motif in binding core position 4 of the other protection-associated alleles. Similarly, DRB1*01:01 displays a SAG motif instead of the SANG motif seen across the rest of the protection-associated alleles. This lack of asparagine binding in the binding core of DRB1*01:01 maps to the P11L amino acid substitution.

Although position 1 of the protection-associated DRB1 alleles shares a consistent hydrophobic motif, this residue is found across all risk-associated DRB1 alleles as well. This finding potentially reflects the previously reported limited binding repertoire of the HLA-DRB1 pocket 1 position, generally enabling the binding of the following amino acids: FYLVM [48]. Moreover, the occurrence of the FY amino acids in the binding cores perfectly correlates with the G86V dimorphism in the HLA-DRB1 sequence.

As such, our data suggest that, at least for DRB1, protection-associated class II HLA alleles have similar binding core properties that discern them from risk-associated HLA alleles.

### 3.5. HLA Class II Epitope Conservation across Leishmania Species and HLA Allotypes

The finding that protection-associated HLA-DRB1 alleles are consistently associated with leishmaniasis, and share common binding motifs, highlights that epitopes that bind to these HLA alleles are the most likely to confer a disease-specific immune response. Thus, these epitopes may be attractive candidates for inclusion in future epitope discovery studies.

To further prioritize the in-silico predicted epitopes for future discovery, we set out to identify whether any epitopes were conserved across *Leishmania* spp. (Figure 3), and whether these epitopes bind across the different protection-associated HLA alleles (Figure 4 and Figure 5).

A total of 44 protection-associated strong-binding epitopes are conserved across *L. braziliensis*, *L. donovani*, *L. infantum,* and *L. mexicana*. These epitopes and their properties are listed in an excel file provided at https://github.com/BioinformaNicks/LeisHLA (accessed 17/02/2021).

Lastly, Table 3 lists the strong-binding epitopes conserved across species that overlap with those that bind across different alleles. A total of 14 strong-binding epitopes are conserved across all species and bound across at least two different HLA allotypes (range 2–4).

## 4. Discussion

We carried out the first multi-species and proteome-wide T cell epitope binding predictions for leishmaniasis-associated HLA alleles (six protection-associated and 11 risk-associated) and further studied the underlying properties of HLA-DRB1 alleles associated with susceptibility or resistance to infection. We demonstrated that several functionally important amino acid variants may differentiate protection- from risk-associated HLA-DRB1 alleles. This further translated in common epitope binding motifs across the protection-associated alleles, altering the epitope binding repertoire, in sharp contrast to the risk-associated HLA profile which displayed a promiscuous nature. Finally, this knowledge was used to identify species-conserved and HLA-unrestricted binding epitopes across the vast amount of possible epitopes. This prioritized list of epitopes can serve as guidance for the study of *Leishmania*-specific immunity and future epitope discovery and may drive the development of novel subunit, multi-epitope vaccines.

To explore the mechanisms underpinning the HLA associations in leishmaniasis, we identified several amino acid variants that were shared entirely across either the risk- or the protection-associated HLA-DRB1 alleles included in this study. These amino acid variants are thought to confer distinct (physicochemical and structural) properties on functionally important positions in the HLA-DRB1 structure. This may ultimately impact the disease association status by altered antigen presentation. Similarly, although related to auto-immunity, three amino acid variants located in the binding groove of the HLA-DRB1 locus are known to mediate disease association in rheumatoid arthritis [50]. For leishmaniasis, the identified amino acid variants include residues 9, 37, and 140, which fully differentiate between the leishmaniasis protection- and risk-associated HLA alleles, but also residues 10–13, 96, 133, 142, 149, and 233 which were shared across the majority of the alleles within the association groups but not shared in one allele. In general, DRB1*04:07 and DRB1*01:01 were the alleles for which these exceptions were noted. We observed some degree of influence between the epitope binding pocket positions, where the binding of a specific amino acid at a particular binding pocket position alters the range of amino acids able to bind at another position [48]. DRB1*01:01 and DRB1*04:07 both exhibit several distinct mutations in binding pocket positions that may be permissive for variants at other positions. Residues 9 and 37 of the HLA-DRB1 sequence dictate the strength and stability of binding of the ninth amino acid of an epitope to the epitope binding pocket of the MHC molecule [23]. Residue 37 is the main contributor in shaping the electrostatic properties of position 9 of the epitope binding pocket [51]. At this residue, asparagine induces a positive charge while tyrosine induces a negative charge, ultimately restricting the range of distinct epitopes that are able to bind this pocket [51]. Residue 140 interacts with the CD4 co-receptor, and a hydrophobic alanine is shared across the protection-associated alleles while the risk-associated alleles share a hydrophilic threonine. This indicates that these structural and physicochemical variations may alter the epitope binding repertoire and explain the disease association status [52].

Earlier work of our lab on the Varicella-Zoster virus (VZV) demonstrated that HLA class I alleles associated with risk for postherpetic neuralgia demonstrated a significantly lower affinity for VZV-derived epitopes than those associated with lower risk [17]. Thus, we postulated that the structural and physicochemical variations identified in leishmaniasis-associated HLA alleles would alter their binding affinity. Yet, the epitope binding affinity distributions were not significantly different between protection- and risk-associated HLA alleles in leishmaniasis. However, we employed leishmaniasis-wide relative affinity comparisons, treating all *Leishmania*-derived epitopes as equals, while studies for other pathogens have shown that disease susceptibility is more directly associated with epitope-affinity within specific timely-expressed proteins rather than the full proteome [53]. Consequentially, the effect of the HLA disease-association on the binding affinity of (not yet known) leishmaniasis epitopes, that play a key role in the immune response, may be masked by predicted epitopes that are not functionally important. However, this cannot yet be verified due to a lack of well-characterized immune response proteins and derived epitopes.

We confirmed a common combination of binding motifs in the binding cores of predicted epitopes that bind to the known protection-associated HLA alleles. These common binding motifs at positions 4, 6, and 9 of the binding core map to several of the identified variants in the amino acid sequence of the HLA-DRB1 alleles. In a prior attempt to identify the effect of leishmaniasis-associated HLA variants on antigen presentation, Singh et al. characterized the binding pocket preferences of one protection-associated allele and one susceptibility-associated allele for *L. donovani*-derived epitopes of a subset of immunogenic proteins [40]. Similar to our findings, they demonstrate that the binding pocket positions 4 and 6 of the protection-associated HLA-DRB1 allele exhibit preference for hydrophobic and polar amino acids. In contrast, the binding pocket positions of the risk-associated HLA-DRB1 allele preferred basic amino acids and were generally more promiscuous. To minimize the confounding effect of natural variation in the highly polymorphic HLA locus, we successfully expanded on their findings in only two disease-associated HLA alleles, replicating it across ten different disease-associated HLA-DRB1 alleles and whole proteomes of multiple *Leishmania* species.

Additionally, Singh et al. demonstrated that in vitro stimulation of whole blood-derived from cured patients, homozygous for the disease-associated HLA alleles, with peptides captured from the protection-associated HLA-DRB1*15:01 allele resulted in a higher IFN-γ to IL-10 ratio response than those captured from the risk-associated HLA-DRB1*13:01 allele. IFN-γ is linked to increased parasite clearance through increased parasitotoxic nitrate production [6,54]. This finding could indicate that preferential binding of an epitope to a protection- or risk-associated allele can elicit a protection- or risk-associated T cell response. This means disease-associated epitopes may be directly wheeled for generating insight in functional *Leishmania*-specific T cell immune responses, and to guide the immunological evaluation of these T cell responses in future vaccine trials.

Our unbiased epitope prediction approach used stringent filter criteria to identify only those epitopes that are likely to get presented in vivo. This approach still yielded several epitopes from antigens known to be immunogenic (at least in animal models), including, for example, amastin, kinesin, 60S ribosomal subunit L31, GP63, and LeIF (see Appendix A) [55,56,57,58,59,60]. Our epitope candidate prioritization method, which placed preference on conservancy across species and promiscuous binding across alleles, resulted in several epitopes of antigens we speculate to be immunogenic in humans. More specifically, several peptides derived from LeIF2A were predicted to be both conserved across all species and to bind across different protection-associated HLA alleles. Of note, the LeIF variant commonly thought to be immunogenic in animals is typically LeIF4A instead of LeIF2A. Moreover, an epitope of Ribosomal protein S3 was predicted to be conserved and promiscuous as well. This antigen was shown to be involved in the Th1/Th2 skewing of the immune response in a BALB/c mice model [61]. Thus, we postulate that with the in silico predicted and prioritized epitopes, generated by this study, we provide an important resource for future applications in rational vaccine discovery. Before these epitopes are included for further vaccine candidate prioritization, broader promiscuous binding than to protection-associated HLA alleles needs to be tested to acquire a wide population coverage. It is not known whether any possible protective effect elicited by a protection-associated epitope is fully restricted to protection-associated HLA alleles, or if these epitopes elicit an immunodominant response in protection-associated HLA alleles and a subdominant response in non-disease associated allotypes. In addition, the subset of the population that requires protection by means of induced immunity through vaccination may differ from those that carry protection-associated HLA allotypes. In this case, a multi-epitope vaccine including multiple subdominant epitopes may be enough to elicit protective effects in a broader population.

In this work, we demonstrated a feasible approach for generating insight into the mechanisms that contribute to HLA disease-associations. However, this approach is heavily dependent on the quality of prior HLA association studies. The HLA association studies we included are limited in number and vary considerably in species studied, study age, statistical power, and HLA typing methods. HLA typing methods used in earlier studies include sequence-specific oligonucleotide- or primer polymerase chain reaction and lymphocytotoxicity assays. These are often restricted to either MHC class I or class II and are generally low in resolution, enabling typing up to only the allele group level [62].

Nonetheless, the association of HLA-DRB1 with leishmaniasis susceptibility was robust. Variants of the DRB1*15 and DRB1*16 allele groups were associated with lower susceptibility for leishmaniasis in all but one study, regardless of HLA-II typing method. Other variants of the HLA-DRB1 locus were not consistently associated with (lower or higher) susceptibility to leishmaniasis. However, the reasons for this may be multifactorial. Firstly, HLA allele frequencies vary considerably across the different study populations. Secondly, most of the included studies had low sample sizes. Hence, they may have insufficient power to uncover associations with more rare alleles in the study population or alleles that are less strongly associated with leishmaniasis. We argue that the limitations of HLA typing methods across the included studies do not invalidate the robust association of the HLA-DRB1 locus with leishmaniasis, their inclusion in our study and the subsequent results of our work. It does mean that there might be other, yet unknown, leishmaniasis-associated HLA alleles that we could not include in our analysis, HLA-DRB1 or otherwise, which could have led to additional insights.

It is also worth mentioning that most of these HLA association studies were restricted to the South American continent. Little data has been generated for other endemic regions such as the African continent. Future endeavors that cover this research gap are warranted. Nevertheless, despite the small number of HLA association studies of variable quality conducted to date, we were still able to generate valuable knowledge on the antigen presentation in leishmaniasis.

## 5. Conclusions

The data presented in this study strongly suggest that polymorphism in the HLA-DRB1 region, and the resulting variation in the epitope binding pocket, mediates the association between HLA-DRB1 and leishmaniasis disease outcome. Protection-associated HLA alleles display common epitope binding motifs, and these motifs map to amino acid substitutions largely or fully shared across these alleles. This knowledge can be used to support future evaluations of epitope-specific immune responses in *Leishmania*. Moreover, our in silico prioritization of potential T cell epitopes can be directly wheeled as a resource for future reverse-vaccinology approaches, guiding the development of novel subunit multi-epitope vaccines.

## Figures and Tables

**Figure 1 vaccines-09-00270-f001:**
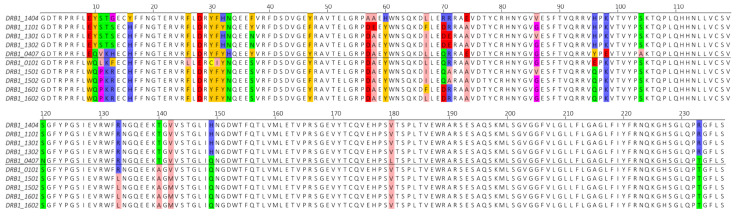
A multiple sequence alignment of the risk-associated HLA-DRB1 alleles (top 5) versus the protection-associated HLA-DRB1 alleles (bottom 5). Non-similar positions were colored using the physicochemical properties of the respective amino acids. Rose = hydrophobic/aliphatic; Orange = aromatic; Blue = positively charged; Red = negatively charged; Green = hydrophilic; Purple = conformationally special; Yellow = Cysteine.

**Figure 2 vaccines-09-00270-f002:**
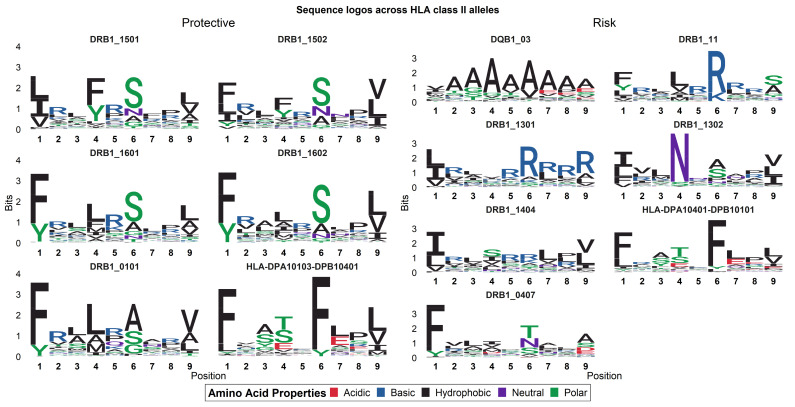
Consensus sequence logos of the 9 mer binding cores of predicted epitopes across HLA class II alleles. This plot shows the sequence conservation level (in bits) of an amino acid occurring at a certain position in the binding core. Protection and risk-associated HLA alleles have been separated in a 2-column matrix.

**Figure 3 vaccines-09-00270-f003:**
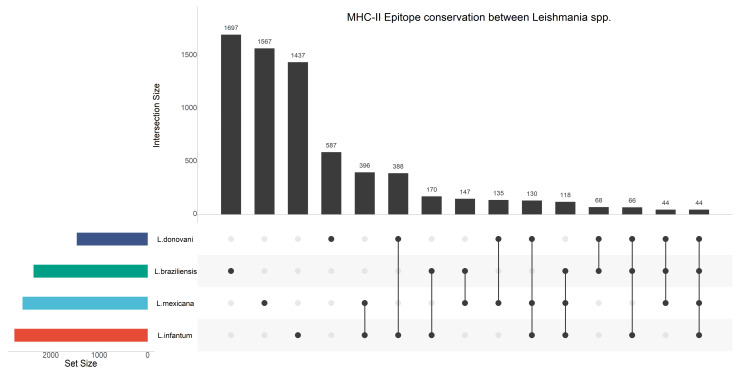
Upset Plot of strong-binding epitopes that are 1) unique to protection-associated alleles and 2) conserved across two major VL- (*L. infantum* and *L. donovani*) and two major CL-causing *Leishmania* species (*L. braziliensis* and *L. mexicana*). The set size denotes the total number of strong-binding epitopes predicted for each species, while the intersection size denotes the number of strong-binding epitopes shared across the species.

**Figure 4 vaccines-09-00270-f004:**
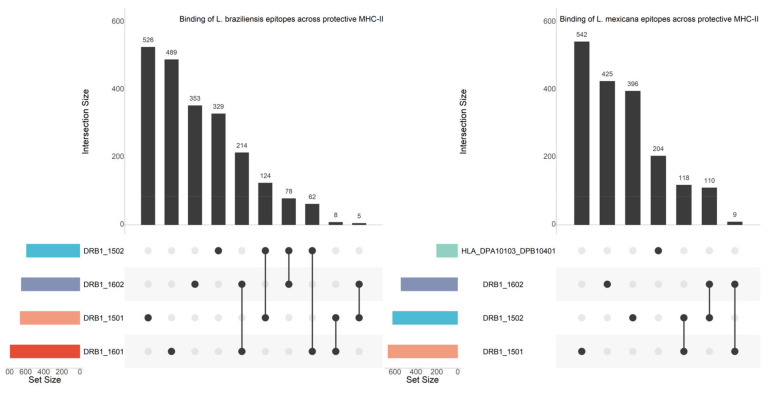
Upset Plots showing unique strong-binding epitopes of the CL-causing species (*L. braziliensis* and *L. mexicana*) that bind across the protection-associated alleles known for these species. The set size denotes the total number of strong-binding epitopes predicted for each species, while the intersection size denotes the number of strong-binding epitopes shared across the species.

**Figure 5 vaccines-09-00270-f005:**
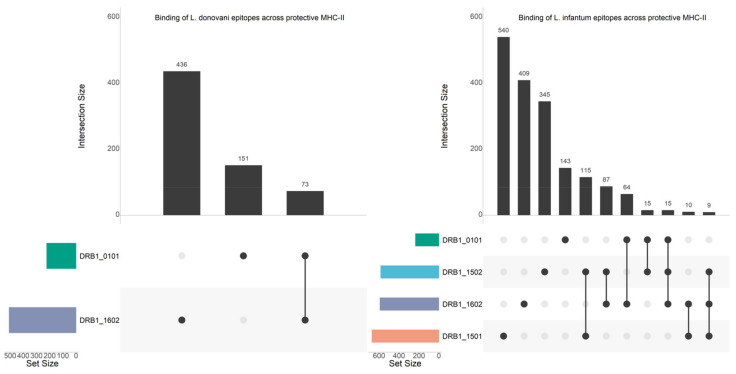
Upset Plots showing unique strong-binding epitopes of the VL-causing species (*L. donovani* and *L. infantum*) that bind across the protection-associated alleles known for these species. The set size denotes the total number of strong-binding epitopes predicted for each species, while the intersection size denotes the number of strong-binding epitopes shared across the species.

**Table 1 vaccines-09-00270-t001:** Leishmaniasis-associated HLA alleles extracted from the literature were divided into those that confer protection against contracting the disease (Protection) and those that increase the susceptibility for the disease (Risk). CL: Cutaneous leishmaniasis; LCL: Localized cutaneous leishmaniasis; DCL: Diffuse cutaneous leishmaniasis; MCL: Mucocutaneous leishmaniasis; VL: Visceral leishmaniasis. ^†^ = *L. donovani* does not typically cause CL, but this species has been reported to cause both VL and CL in the Sri Lankan population [39]. * = Included patients with both LCL and DCL.

Type	Reference	Protective Alleles	Risk Alleles	Sample Size	Species	Study Location
VL	Singh et al., 2018 [40]	DRB1*0101 DRB1*1501 DRB1*1502 DRB1*1602	DRB1*11 DRB1*1404 DRB1*1301 DRB1*1302	*Patients N = 889* *Control N = 977*	*L. donovani* *L. infantum*	India Brazil
VL	Fakiola et al., 2013 [36]	DRB1*01 DRB1*15 DRB1*16	DRB1*11 DRB1*13 DRB1*14	*Three cohorts.* *Total Patients N = 2287* *Total Control N = 3692*	*L. donovani* *L. infantum*	India Brazil
VL	Faghiri et al., 1995 [41]		A*26	*Patients N = 52* *Control N = 222*	*L. donovani*	Iran
MCL	Petz-Erler et al., 1991 [42]	DRB1*15 DRB1*16	DQB1*03	*Patients N = 43* *Control N = 111*	*L. braziliensis*	Brazil
LCL ^†^	Samaranayake et al., 2016 [12]	DRB1*15-DQB1*06	DRB1*15	*Patients N = 140* *Control N = 140*	*L. donovani* ^†^	Sri Lanka ^†^
LCL	Ribas-Silva et al., 2015 [43]		C*04	*Patients N = 186* *Control N = 278*	*L. braziliensis*	Brazil
LCL	Olivo-Díaz et al., 2004 [44]	DRB1*15 DRB1*16 DPB1*0401	DRB1*0407 DQA1*3011 DPA1*0401 DPB1*0101 DRB1*0407-DQA1*3011-DQB1*0302 DRB1*0407-DQA1*3011-DQB1*0301	*Patients N = 65* *Control N = 100*	*L. mexicana*	Mexico
CL *	El-Mogy et al., 1993 [45]	-	A*11 B*05 B*07	*Patients N = 27* *Control N = 200*	*L. major*	Egypt
LCL	Lara et al., 1991 [37]	B*15	A*28 Bw22 DQw3 Bw22-CF31 Bw22-DR*11-DQ*7	*24 families.* *Patients N = 127* *Control N = 160*	*L. guyanensis* *L. braziliensis*	Venezuela

**Table 2 vaccines-09-00270-t002:** Functionally important observed amino acid substitutions are shown between protection-associated (top 5, white) and risk-associated (bottom 5, greyscale) HLA alleles. Bolded residues denote amino acid variants shared across several alleles, and variants shared across remaining other alleles are underlined. The respective epitope binding pocket positions are denoted with Px. A = Alanine; C = Cysteine; D = Aspartic acid; E = Glutamic acid; F = Phenylalanine; G = Glycine; H = Histidine; I = Isoleucine; K = Lysine; L = Leucine; M = Methionine; N = Asparagine; P = Proline; Q = Glutamine; R = Arginine; S = Serine; T= Threonine; V = Valine; W = Tryptophan; Y = Tyrosine.

Sequence Residue	9	10	11	12	13	26	28	30	37	67	70	71	74	86	96	133	140	142	149	233
DRB1*15:01	W	Q	P	K	R	F	D	Y	S	I	Q	A	A	V	Q	L	A	V	Q	T
DRB1*15:02	W	Q	P	K	R	F	D	Y	S	I	Q	A	A	G	Q	L	A	V	Q	T
DRB1*16:01	W	Q	P	K	R	F	D	Y	S	*F*	D	R	A	G	Q	L	A	V	Q	T
DRB1*16:02	W	Q	P	K	R	F	D	Y	S	L	D	R	A	G	Q	L	A	V	Q	T
DRB1*01:01	W	Q	L	K	F	L	E	C	S	L	Q	R	A	G	E	R	A	V	Q	T
DRB1*04:07	E	Q	V	K	H	F	D	Y	Y	L	Q	R	E	G	Y	R	T	V	Q	T
DRB1*11:01	E	Y	S	T	S	F	D	Y	Y	*F*	D	R	A	G	H	R	T	M	H	R
DRB1*13:01	E	Y	S	T	S	F	D	Y	N	I	D	*E*	A	V	H	R	T	M	H	R
DRB1*13:02	E	Y	S	T	S	F	D	Y	N	I	D	*E*	A	G	H	R	T	M	H	R
DRB1*14:04	E	Y	S	T	G	F	D	Y	F	L	R	R	E	V	H	R	T	M	H	R
Functional position	P9		P6		P4	P4	P4/7	P6	P9	P7/TCR	P4	P4/7	P4	P1		CD4 contact	CD4 contact	CD4 contact	CD4 contact	

**Table 3 vaccines-09-00270-t003:** Prioritized in silico predicted strong-binding epitopes that are (A) conserved across the included species and (B) bind across different HLA alleles. Ranked by the number of HLA alleles and the protein these epitopes are derived from.

HLA Alleles	Peptide	Binding Core	Protein
DRB1*15:01;DRB1*15:02;DRB1*16:01;DRB1*16:02	QTFTLFKSLRAHMLP	FTLFKSLRA	Eukaryotic translation initiation factor eIF2A
DRB1*16:01;DRB1*01:01;DRB1*16:02	TFTLFKSLRAHMLPL	FKSLRAHML	Eukaryotic translation initiation factor eIF2A
DRB1*16:01;DRB1*01:01;DRB1*16:02	FTLFKSLRAHMLPLT	FKSLRAHML	Eukaryotic translation initiation factor eIF2A
DRB1*16:01;DRB1*01:01;DRB1*16:02	HFTSYRHLPALRLLS	YRHLPALRL	hypothetical protein, conserved
DRB1*16:01;DRB1*01:01;DRB1*16:02	FTSYRHLPALRLLSA	YRHLPALRL	hypothetical protein, conserved
DRB1*16:01;DRB1*01:01;DRB1*16:02	NDEFHMLRSASIKII	FHMLRSASI	Carbamoyl-phosphate synthase small chain
DRB1*16:01;DRB1*01:01;DRB1*16:02	HRLFILLHGQPIAQS	FILLHGQPI	Ankyrin repeats
DRB1*16:01;DRB1*01:01;DRB1*16:02	RLFILLHGQPIAQSP	FILLHGQPI	Ankyrin repeats
DRB1*15:02;DRB1*01:01;DRB1*16:02	TNAFRLQLSNPIIFS	FRLQLSNPI	Ring finger domain containing protein
DRB1*15:02;DRB1*16:02	NPTNAFRLQLSNPII	FRLQLSNPI	Ring finger domain containing protein
DRB1*15:02;DRB1*16:02	PTNAFRLQLSNPIIF	FRLQLSNPI	Ring finger domain containing protein
DRB1*15:02;DRB1*16:02	NPGYISLFSTPIVKI	ISLFSTPIV	topoisomerase IV, subunit A
DRB1*15:02;DRB1*16:02	FIILPFIFIPSNTIS	FIFIPSNTI	ABC-2 family transporter protein
DRB1*15:02;DRB1*16:02	DRIFRSFRTNNVKMT	FRSFRTNNV	hypothetical protein, conserved

## Data Availability

Data and code are provided at a Github repository for this project: https://github.com/BioinformaNicks/LeisHLA (accessed 17/02/2021).

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
