# Peer review of "HLA-DRB1 Alleles Associated with Lower Leishmaniasis Susceptibility Share Common Amino Acid Polymorphisms and Epitope Binding Repertoires"

_vaccines, 2021, doi:10.3390/vaccines9030270_

Round 1
Reviewer 1 Report
In this manuscript, Vrij et al present an analysis of the T cell epitope binding ability of peptides in a variety of Leishmania species, covering their entire proteome. This gave the information on HLA alleles that are associated with leishmaniasis. Using this, leishmaniasis protection or risk-associated alleles are distinguished based on the amino acid differences, and their binding epitopes are predicted. However, they observed no significant difference in the binding affinity for protection or risk-associated alleles. However, they found distinct amino acid substitution profiles in the class II HLA-DRB1 alleles. Further, they performed in silico screening for conserved peptides in the Leishmania genus that bind to those protection-associated HLA alleles. Their study is relevant in two ways. One, it brings out the importance of HLA-DRB1 polymorphism in Leishmania disease manifestation, and two, this study could be a useful dataset in assessing the immune response in Leishmania infection and will possibly help in vaccine development.
Comments:
-This is a well-researched and well-written article. My major criticism lies in the very base of the paper, that is the literature this paper is based on. I assume the HLA typing methodologies may not be uniform across these papers and that brings issues. Generating new HLA data across various populations is beyond the scope of the paper, but authors should discuss this caveat elaborately in the discussion.
-The title is non-intuitive and difficult to comprehend. Authors should consider re-framing it.
-The authors often use protective-associated. It should be protection-associated.
Author Response
We would like to thank the reviewer for the constructive feedback. We have addressed all points below.
In this manuscript, Vrij et al present an analysis of the T cell epitope binding ability of peptides in a variety of Leishmania species, covering their entire proteome. This gave the information on HLA alleles that are associated with leishmaniasis. Using this, leishmaniasis protection or risk-associated alleles are distinguished based on the amino acid differences, and their binding epitopes are predicted. However, they observed no significant difference in the binding affinity for protection or risk-associated alleles. However, they found distinct amino acid substitution profiles in the class II HLA-DRB1 alleles. Further, they performed in silico screening for conserved peptides in the Leishmania genus that bind to those protection-associated HLA alleles. Their study is relevant in two ways. One, it brings out the importance of HLA-DRB1 polymorphism in Leishmania disease manifestation, and two, this study could be a useful dataset in assessing the immune response in Leishmania infection and will possibly help in vaccine development.
Comments:
-This is a well-researched and well-written article. My major criticism lies in the very base of the paper, that is the literature this paper is based on. I assume the HLA typing methodologies may not be uniform across these papers and that brings issues. Generating new HLA data across various populations is beyond the scope of the paper, but authors should discuss this caveat elaborately in the discussion.
We thank the reviewer for this suggestion. We were indeed aware of this caveat and the manuscript already briefly touched upon it in the discussion. However, we fully agree that due to its importance and impact, we should have further elaborated upon it. Accordingly, we have now significantly expanded this section and discuss in detail the impact of this limitation.
Specifically, we modified the section (L532):
Although we demonstrate a feasible approach for generating insight in the mechanisms that contribute to HLA disease-associations, this approach is heavily de-pendent on the quality of prior HLA association studies. The HLA association studies included in this work are limited in number, and vary considerably in species studied, study age, statistical power and methods of HLA typing. The methods of HLA typing used in the earlier included studies are often restricted to either class I or class II, and are low resolution enabling only typing up to the level of the allele group [63].
To (L539):
In this work, we demonstrated a feasible approach for generating insight into the mechanisms that contribute to HLA disease-associations. However, this approach is heavily dependent on the quality of prior HLA association studies. The HLA association studies we included are limited in number and vary considerably in species studied, study age, statistical power, and HLA typing methods. HLA typing methods used in earlier studies include sequence-specific oligonucleotide- or primer polymerase chain reaction and lymphocytotoxicity assays. These are often restricted to either MHC class I or class II and are generally low in resolution, enabling typing up to only the allele group level [63].
Nonetheless, the association of HLA-DRB1 with leishmaniasis susceptibility was robust. Variants of the DRB1*15 and DRB1*16 allele groups were associated with lower susceptibility for leishmaniasis in all but one study, regardless of HLA-II typing meth-od. Other variants of the HLA-DRB1 locus were not consistently associated with (low-er or higher) susceptibility to leishmaniasis. However, the reasons for this may be multifactorial. Firstly, HLA allele frequencies vary considerably across the different study populations. Secondly, most of the included studies had low sample sizes. Hence, they may have insufficient power to uncover associations with more rare alleles in the study population or alleles that are less strongly associated with leishmaniasis. We argue that the limitations of HLA typing methods across the included studies do not in-validate the robust association of the HLA-DRB1 locus with leishmaniasis, their inclusion in our study and the subsequent results of our work. It does mean that there might be other, yet unknown, leishmaniasis-associated HLA alleles that we could not include in our analysis, HLA-DRB1 or otherwise, which could have led to additional insights.
-The title is non-intuitive and difficult to comprehend. Authors should consider re-framing it.
We thank the reviewer for this comment and agree. We have renamed and simplified the title to “HLA-DRB1 alleles associated with lower leishmaniasis susceptibility share common amino acid polymorphisms and epitope binding repertoires”.
-The authors often use protective-associated. It should be protection-associated.
We thank the reviewer for pointing this out. We have changed ‘protective-associated’ to ‘protection-associated’ throughout the manuscript.
Reviewer 2 Report
Review of Vaccines 1135430:
This is a very well written and well documented manuscript on a topic of extreme interest to global health, vaccinology, and neglected tropical diseases. The work is well organized, the pertinent literature is cited and discussed, the illustrations and tables are clear and make the findings of this research more clear. This is an unusual approach, especially for parasitic diseases which tend to be less frequent objects for vaccine development. The authors have done a good job of combining analytical literature search with genetic database analysis to come up with new patterns that are revealing of new possible relationships and potential applications.
One minor correction that needs to be made is an entomological technicality. According to entomological terminology conventions, a member of the order Diptera is spelled as two words. Thus, the vectors of leishmaniasis are sand flies (not sandflies). Spelling an insect name as one word is reserved for members of other orders (e.g., butterflies (Lepidoptera), dragonflies (Odonata), mayflies (Ephemeroptera), stoneflies (Plecoptera), etc.). This misspelling should be corrected in the text of this manuscript.
In my mind, the abstract could be modified slightly to better explain the way the study was designed. This important point is very clear in reading the Materials and Methods, but it does not emerge as clearly in the abstract. I recommend that the authors consider this.
Following very minor revisions as outlined above, I consider this manuscript to be well worthy of publication.
Author Response
We would like to thank the reviewer for the constructive feedback. We have addressed all points below.
This is a very well written and well documented manuscript on a topic of extreme interest to global health, vaccinology, and neglected tropical diseases. The work is well organized, the pertinent literature is cited and discussed, the illustrations and tables are clear and make the findings of this research more clear. This is an unusual approach, especially for parasitic diseases which tend to be less frequent objects for vaccine development. The authors have done a good job of combining analytical literature search with genetic database analysis to come up with new patterns that are revealing of new possible relationships and potential applications.
One minor correction that needs to be made is an entomological technicality. According to entomological terminology conventions, a member of the order Diptera is spelled as two words. Thus, the vectors of leishmaniasis are sand flies (not sandflies). Spelling an insect name as one word is reserved for members of other orders (e.g., butterflies (Lepidoptera), dragonflies (Odonata), mayflies (Ephemeroptera), stoneflies (Plecoptera), etc.). This misspelling should be corrected in the text of this manuscript.
We thank the reviewer for pointing out this convention. We have changed ‘sandflies’ to ‘sand flies’ throughout the manuscript.
In my mind, the abstract could be modified slightly to better explain the way the study was designed. This important point is very clear in reading the Materials and Methods, but it does not emerge as clearly in the abstract. I recommend that the authors consider this.
We thank the reviewer for this comment. We have extensively modified the abstract to better reflect the design and methodology of the study.
Following very minor revisions as outlined above, I consider this manuscript to be well worthy of publication.
Reviewer 3 Report
Leishmaniasis is an endemic spanning multiple continent and classified as neglected tropical diseases. Thus, it’s very important to know the mechanism of host-pathogen interaction to develop new therapeutics and effective vaccines. The authors used immunoinformatic tools to investigate the mechanism of host-pathogen interaction. Previously, various human leukocyte antigen (HLA) gene clusters are identified as genetic susceptibility factors for leishmaniasis. In the current manuscript, the authors identified several amino acid polymorphism (in the HLA gene sequence) which distinguishes protective HLA-DRB1 allels from risk-associated one. Their data show that the polymorphism in the HLA gene sequence results in variation epitope binding pocket which finally determines the outcome of leishmaniasis disease upon infection. The current work is important to understand host-pathogen interaction in leishmaniasis and will provide necessary foundation to develop potential multi-epitope-based vaccines.
The authors provided sufficient background. The research questions are relevant and meaningful. Data were analyzed properly and interpreted appropriately. The conclusion was supported by the data. The methods described in the manuscript is sufficient to follow. Below are some minor comments,
Minor comments:
- Line 99: “… presentation..” -> “… presentation.”
- Line 102-103: “… to A) explore the properties underpinning the known class I 102 and class II HLA associations …” Class I and Class II HLA associations are introduced here for the first time. Please introduce/discuss class I and class II HLA associations before.
Author Response
We would like to thank the reviewer for the constructive feedback. We have addressed all points below.
Comments and Suggestions for Authors
Leishmaniasis is an endemic spanning multiple continent and classified as neglected tropical diseases. Thus, it’s very important to know the mechanism of host-pathogen interaction to develop new therapeutics and effective vaccines. The authors used immunoinformatic tools to investigate the mechanism of host-pathogen interaction. Previously, various human leukocyte antigen (HLA) gene clusters are identified as genetic susceptibility factors for leishmaniasis. In the current manuscript, the authors identified several amino acid polymorphism (in the HLA gene sequence) which distinguishes protective HLA-DRB1 allels from risk-associated one. Their data show that the polymorphism in the HLA gene sequence results in variation epitope binding pocket which finally determines the outcome of leishmaniasis disease upon infection. The current work is important to understand host-pathogen interaction in leishmaniasis and will provide necessary foundation to develop potential multi-epitope-based vaccines.
The authors provided sufficient background. The research questions are relevant and meaningful. Data were analyzed properly and interpreted appropriately. The conclusion was supported by the data. The methods described in the manuscript is sufficient to follow. Below are some minor comments,
Minor comments:
Line 99: “… presentation..” -> “… presentation.”
We thank the reviewer for spotting this typo and have updated the manuscript accordingly.
Line 102-103: “… to A) explore the properties underpinning the known class I 102 and class II HLA associations …” Class I and Class II HLA associations are introduced here for the first time. Please introduce/discuss class I and class II HLA associations before.
We thank the reviewer for this suggestion and have now introduced class I and class II HLA associations earlier.
More specifically we changed:
L58: Several major histocompatibility complex (MHC) class I and class II genes of the human leukocyte antigen (HLA) gene cluster have been identified as genetic susceptibility factors for leishmaniasis, with variants affecting disease outcome both positively and negatively [11, 12].
To L62:
Several major histocompatibility complex (MHC) class I and class II genes of the human leukocyte antigen (HLA) gene cluster have been identified as genetic susceptibility factors for leishmaniasis, with variants affecting disease outcome both positively and negatively [11, 12].
And we changed L64:
The varying HLA genes code for distinct MHC glycoproteins, divided into class I and class II, and these MHC molecules present antigens to CD8+ and CD4+ T cells, respectively, eliciting a T cell-mediated immune response upon activation.
To L68:
The varying HLA genes code for distinct MHC glycoproteins, divided into class I and class II, and these MHC molecules present antigens to CD8+ and CD4+ T cells, respectively, eliciting a T cell-mediated immune response upon activation.